# (R)Evolution in Allergic Rhinitis Add-On Therapy: From Probiotics to Postbiotics and Parabiotics

**DOI:** 10.3390/jcm11175154

**Published:** 2022-08-31

**Authors:** Martina Capponi, Alessandra Gori, Giovanna De Castro, Giorgio Ciprandi, Caterina Anania, Giulia Brindisi, Mariangela Tosca, Bianca Laura Cinicola, Alessandra Salvatori, Lorenzo Loffredo, Alberto Spalice, Anna Maria Zicari

**Affiliations:** 1Department of Maternal Infantile and Urological Sciences, Sapienza University of Rome, 00185 Rome, Italy; 2Allergy Clinic, Casa di Cura Villa Serena, 65013 Genoa, Italy; 3Department of Translational and Precision Medicine, Sapienza University of Rome, 00185 Rome, Italy; 4Allergy Center, IRCCCS Ist. Giannina Gaslini, 65013 Genoa, Italy; 5Department of Molecular Medicine, Sapienza University of Rome, 00185 Rome, Italy; 6Department of Woman and Child of General and Specialistic Surgery, University of Campania “Luigi Vanvitelli”, 80138 Napoli, Italy; 7Department of Internal Medicine and Medical Specialties, Sapienza University, 00185 Rome, Italy; 8Child Neurology Division, Department of Pediatrics, Sapienza University of Rome, 00185 Rome, Italy

**Keywords:** probiotics, postbiotics, parabiotic, allergic rhinitis, immune system, microbiota, atopic march, allergic disease, allergic inflammation

## Abstract

Starting from the “Hygiene Hypothesis” to the “Microflora hypothesis” we provided an overview of the symbiotic and dynamic equilibrium between microbiota and the immune system, focusing on the role of dysbiosis in atopic march, particularly on allergic rhinitis. The advent of deep sequencing technologies and metabolomics allowed us to better characterize the microbiota diversity between individuals and body sites. Each body site, with its own specific environmental niches, shapes the microbiota conditioning colonization and its metabolic functionalities. The analysis of the metabolic pathways provides a mechanistic explanation of the remote mode of communication with systems, organs, and microflora of other body sites, including the ecosystem of the upper respiratory tract. This axis may have a role in the development of respiratory allergic disease. Notably, the microbiota is significant in the development and maintenance of barrier function; influences hematopoiesis and innate immunity; and shows its critical roles in Th1, Th2, and Treg production, which are necessary to maintain immunological balance and promote tolerance, taking part in every single step of the inflammatory cascade. These are microbial biotherapy foundations, starting from probiotics up to postbiotics and parabiotics, in a still-ongoing process. When considering the various determinants that can shape microbiota, there are several factors to consider: genetic factors, environment, mode of delivery, exposure to antibiotics, and other allergy-unrelated diseases. These factors hinder the engraftment of probiotic strains but may be upgradable with postbiotic and parabiotic administration directly on molecular targets. Supplementation with postbiotics and parabiotics could represent a very exciting perspective of treatment, bypassing probiotic limitations. At present, this avenue remains theoretical and to be explored, but it will certainly be a fascinating path to follow.

## 1. Introduction

The human microbiota is a complex and dynamic biological community that colonizes different parts of the body. In particular, the gut microbiota, which has been studied for the longest time, entails about 10 times more bacterial cells than the number of human cells and over 100 times the amount of genomic content (microbiome) compared to the human genome [1,2,3].

Due to the high number of bacterial cells present in the organism, the microbiota can be considered a real organ provided by its self-sustaining physiology and pathology, at the same time influenced by the host’s environmental pressures [4]. Therefore, impairment in one has an impact on the health–disease status of the other. Many human diseases, including type 2-mediated disorders, have these reciprocal influences. In other words, there is bidirectional talk between the microbiota and the human body [5].

In 1989, Strachan proposed the so-called “Hygiene Hypothesis”: reduced environmental exposure to antigenic/infectious sources affects the normal development of the immune system [6]. The “Hygiene Hypothesis” laid the groundwork for “Old Friends” and the “Microflora Hypothesis” [7,8]. The “Microflora Hypothesis” assumes that the biodiversity of gut microbiota plays an important role in shaping host immune development and that derangement (dysbiosis) in the normal gut microbiota composition contributes to the development of immune-mediated disorders, such as allergic diseases [8,9,10].

Moreover, the huge and widespread increase in the prevalence of allergic diseases has led to the coining of a new term: “allergy epidemic” [11]. Namely, the prevalence of allergic diseases has reached 40% of the general population. In this regard, the loss of microbiota biodiversity and humoral immunoregulatory pathways are a consequence of the adoption of the Western lifestyle [12]. Although clear mechanistic insights are still unclear, they constitute one of the most common scientific lines of research to explain the rise in allergy prevalence. Therefore, an additional interventional strategy should probably focus on microbiota rebalance through successful microbial biotherapy. 

## 2. Background: Development of Dynamic Equilibrium

### 2.1. The Microbiota

In the mid-1800s, Professor Theodor Escherich first studied the intestinal microbiota of the infant gut, recognizing its complexity and its pivotal role in gut physiopathology [13]. From Escherich’s forward-looking publications, in less than 200 years, huge advances were made in the knowledge of this articulated ecosystem and its interactions with the human host. To date, the human–microbiome linkage is considered a superorganism, a fruit of evolution, in which both parts develop simultaneously through mutual influences, integrating their functions under genetic and environmental influences [14,15]. As a result of this dynamic and long-lasting equilibrium, the microbiota carries out protective, immunomodulatory, anti-inflammatory, metabolic, neurological, and still unknown functionalities [16].

Microbial colonization starts in early life (as early as the fetal period), and several factors influence its development; first, during and after birth and over the whole life cycle. It is characterized by age-related biodiversity closely depending on health and disease status [16,17]. Fetal–maternal microbiota interchange occurs already during intrauterine life. Many prenatal factors (diet, atopy, smoking status, antibiotic) and the genetic background of the infant may influence the colonization process that occurs in a predetermined order [16,18]. Subsequently, even if the type of delivery is widely considered a significant step in the first microbiota settlement, recently, several studies have proposed a rearrangement of infant microbiota mainly determined by body niches and not by the delivery type. Previous studies have demonstrated that, in adult individuals, not only does each body site have different and unique microbial communities [19,20,21,22] but contiguous sites (e.g., supragingival plaque vs. tongue) also have significant variations in microbiota composition, probably reflecting that even the smallest of variations in body microniche conditions may influence colonization [19,23]. In addition, a study on preterm infants with very low birth weight (LBW) demonstrated that the microbiota of skin, saliva, and stool rapidly diverge (after the first 15 days of life) within the first three weeks of life: the body site was the primary determinant of bacterial community composition in the LBW infants [24]. More recently, an in-depth exploration of infant microbiome maturation highlighted that, at the time of delivery, the microbiota comprises some taxa of the maternal skin and vaginal microbiota, while at 6 weeks of life, the main patterns of community variation are associated with the body site, and the selection is driven by the body niche asset. The infant stool, nares, and oral cavity harbored microbial communities distinct from the parental ones, reflecting age-related physiological differences between the maternal and infant body habitats, including nutrient availability and oxygen exposure [23].

In recent years, the impact of delivery, whether vaginal or cesarean, on the infant microbiota has been subjected to a critical revision because the clinical decision to deliver via cesarean often implies an underlying maternal or fetal morbidity frequently accompanied by medications, including antibiotics and anti-inflammatory and analgesics, all factors that represent significant confounders [25].

In addition, concurrent studies have characterized bacteria mapped to the placenta and amniotic fluid of preterm and healthy term pregnancies in humans [26,27,28,29] and the maternal transmission of bacteria to the fetal gut during gestation in mice, consistent with the microbial colonization of the mouse fetus occurring before delivery [30]. In addition, the meconium has been shown to harbor a microbial community similar to that of the amniotic fluid and placenta [31,32,33] that varies with maternal glycemic control, but not with delivery modality [34]. Thus, even if additional studies are needed to evaluate the potential mechanisms of transmission and their potential impact on fetal microbiota long-term programming, several studies have indicated that the early composition of the gut may be influenced by maternal diet or health status (i.e., pre-pregnancy obesity and maternal glycemic control) in the last trimester of pregnancy, independent of the mode of delivery [35]. Notably, these maternal health conditions also increase the risk that the pregnancy will be delivered by cesarean delivery [36]. Therefore, the predominant role of delivery modality in determining microbiota colonization must be assessed considering the complexity of the clinical setting leading to the choice of delivery mode. 

Breastfeeding provides other influences on microbiota development due to its content of unique oligosaccharides, which act as prebiotics, nondigestible food ingredients stimulating growth and/or activity of probiotics, (e.g., bifidobacterial strains) [16,37]. These findings underscore the likely importance of metabolic pathways when considering the impact of an intervention on microbiota biodiversity. 

### 2.2. The Immune System

At the same time, the proactive development of the immune system (IS) also occurs at the beginning of life, from the fetus stage, and during infancy via specific stages [38,39]. The immune system gradually shifts from a state of maternal–fetal immune tolerance to the training of innate and adaptive immunity according to preset patterns [40]. As soon as we come into the world, immediately, we are subjected to a particular and massive exposure to environment-related antigens, imprinting, via epigenetic changes, our immune homeostasis. Any disturbances interfering with the proper trajectory of immunological development, which is also driven by interactions with personal microbiological heritage, may contribute to enhanced susceptibility to immune-mediated disease [38,39,40].

The microbiota’s significance in immune development during the early stage of life may be long-lasting, creating a “window of opportunity” for proper, durable modulation later in life [41].

### 2.3. The Atopic March

Interestingly, among immune-mediated diseases, the pathogenesis of atopic ones recognizes a typical temporal trend with an early-life onset, initially characterized by atopic dermatitis and food allergy, and afterward by allergic rhinitis and asthma in childhood. This timeline, known as the “atopic march”, is due to a multifactorial-based immunological dysregulation triggering localized manifestations of systemic disease [42].

Notably, the main factors that influence microbiological colonization also condition the development of atopic diseases, including a family history of atopy; the mode of delivery; the type of breastfeeding; diet; antibiotic use; environmental pollutants; and, of course, exposure to allergens (both inhalants and foods) [5]. It is reasonable to suppose that the microbiota, when simultaneously altered by the same factors, play a significant role in inducing (and worsening) allergic disease.

### 2.4. Deviation of the Microbiota/Immune System Axis in Atopy

Thus, the microbiota performs a pivotal role in developing, shaping, and training functionalities of the immune system. The IS controls a well-balanced symbiotic and synergic relationship via cellular and metabolic crosstalk. Therefore, the microbiota–immune system axis can be modulated by acting on the former not only from the gut but also from other body sites (e.g., skin and airways) [43]. 

Indeed, the combination of dysbiosis and predisposition to allergic disease creates a body niche favorable to unbeneficial and pathogenic strains in a vicious cycle perpetuating local and systemic inflammation.

The prevention of allergic rhinitis by breaking down this loop is the dream of allergists [44], and in our opinion, there is a good chance of advancing the knowledge of microbiota metabolites making dreams come true.

## 3. Inflammation in Allergic Rhinitis: Role of Microbiota

### 3.1. Allergic Rhinitis

The term “respiratory allergy” is an umbrella term describing a group of upper and lower airway diseases due to chronic exposure to allergens [45].

Respiratory allergies can coexist within the respiratory tract (e.g., rhinitis and asthma) even though the direct link between them has not been definitively proven [46,47,48]. 

It is accepted that the nose, along with the sinuses, provides 200 cm^2^ of air each day. About 12,000 L is inhaled; a single breath contains millions of different particles, distributed through the respiratory tract [49,50]. In atopic subjects, the breakdown of the respiratory tract immunological barrier results in sensitization [51].

The first injury to respiratory mucosa appears determinant on the release of inflammatory cytokines featuring a cascade effect hesitating in the allergic rhinitis (AR) clinical phenotype [52].

Allergic rhinitis, even if regarded as not a severe disease, accounts for a global burden in terms of its impact on children’s quality of life (sleep quality [53], cognitive function and school attendance [54], and behavioral impairment [55]) and social spending considering the bounded treatment options and the challenge of symptom control. Most of the current treatments are symptomatic (primarily antihistamines and intranasal corticosteroids) [56], except for allergen-specific immunotherapy, to date, the only disease-modifying treatment, even if it is burdened by limited patient compliance, duration, and costs [57,58].

These traditional symptomatic therapies may cause suboptimal disease management with potential side effects in some subjects; thus, they appear to not be in line with the latest advancement of knowledge on disease pathogenesis [59].

Allergic rhinitis features a pathogenic complexity triggered by an improper response to nonpathogenic allergens recognized by the immune system [60].

### 3.2. Microbiota Role in AR Inflammatory Pathways

For several decades, allergic inflammation has been ascribed to T helper 2 (Th2) cells and typified by cytokine production (particularly IL-4, IL-5, IL-9, and IL-13) with a specific cascading effect [61]. Namely, allergic inflammation includes IgE class switching and eosinophil recruitment, infiltration, and activation. Basophils and mast cells maintain inflammation through the release of cytokines, radical oxygen species (ROS), histamine, and leukotrienes. These mediators cause increased vascular permeability, mucous production, smooth muscle hyperreactivity, and remodeling at the “united airways epithelium” [62,63].

Upcoming research suggests the pivotal role of the mucosal epithelial cells, producing and releasing initial key mediators heisting in dendritic cell activation and antigen-specific CD4 + T cell proliferation and differentiation [51].

Thus, innate and adaptive immunity are involved in AR, and the innate lymphoid cells, type 2 (ILC2), are the link between the two arms of immunity [61,64,65]. Those cells reside mainly in mucosal tissues, rapidly producing a large supply of Th2 cytokines without having to recognize antigens, but only as a reaction to an allergenic stimulus that induces IL-33, TSLP, IL-25 (*Alarmin cytokines*), or lipid mediators (PGD2, CysLTs) caused by epithelial cells. ILC2s generate an immediate response within a few hours, while adaptive lymphocytes require days for priming, differentiation, expansion, and chemotaxis. ILCs are organized in niches throughout fetal development and later expand in tissue clusters at birth. Their function phenotypes (ILC1, ILC2, ILC3) differentiate under environmental pressure, including microbiological ones. If the differentiation is biased toward ILC2s during infancy, and by insufficient diversity microbial stimulation, this may lead to allergy. As the manifestation of the Hygiene Hypothesis, early life microbiota may determine the amount of ILC2 in the gut, skin, and airways. Adaptive Th2 lymphocytes gradually replace ILCs in adulthood occupying the same niches [66].

Notably, ILC2s express SCFA (Shorty Chain Fatty Acid)-sensing receptors, and SCFAs, metabolic products, and the main biomarkers of healthy and balanced microbiota have a role in maintaining the optimal number of ILCs in peripheral tissues during infection and inflammatory responses [66,67].

In addition, an imbalance between type 2 immune response and regulatory T cells (Tregs) has a crucial role in the development and chronicity of allergic diseases, including AR [68]. Tregs play a critical role in maintaining immune tolerance to allergens inhibiting type 2 immune cells, such as ILC2, Th2, and IgE-switched B cells. Tregs also enhance tolerogenic dendritic cells, regulatory B cells, and IgG4-switched B cells [5,69].

Dysregulated Tregs and dysbiosis are linked by “Hygiene Hypothesis” models. Increased bacterial or fungal biodiversity observed in rural areas seems to be protective against allergic disorders: specifical bacterial strains (Bacteroides fragilis and clostridium strains) directly influence the development of Tregs. In addition, the fermentation of fibers by gut microbiota leads to the production of SCFAs. *Butylate*, an SCFA, induces Treg production, and *butyrate* supplementation triggers the desensitization of basophils and mast cells and improves Treg functionality in mouse experiments. In addition, SCFAs can promote T cell differentiation into Th1 by counteracting Th1/Th2 imbalance, Th17 effector cells, and Tregs secreting the anti-inflammatory IL-10 and TGF-β cytokines [5,70].

### 3.3. Unexplored Microbiota Diversity

Those above-described events are increasingly characterized models of interplay in the microbiota–immune system, and they illustrate the burden of dysbiosis in influencing allergic inflammation. 

It should be noted that the human gut microbiota is one of the most studied because of its ease in sampling and culture, but microbial ecosystems harbor surfaces that are still poorly characterized. Metagenomic technologies allow us to understand uncultured microbiota, helping us realize the existence of an unexplored diversity of body-wide human microbiomes [71].

This diversity remains uncharacterized, primarily outside of the gut and particularly between populations.

A remarkable study that recovered over 150,000 microbial genomes from ~10,000 human metagenomes spanning 46 datasets from multiple body sites, ages, and geographical origins showed phylogenetic differentiation and a distinctive functional repertoire.

Each of the body sites considered had a clear, distinctive diversity, representative of specific microbial functions and reflecting their age and the Westernization process [72].

This work expanded our point of view about microbiota diversity associated with global populations (non-gut areas and non-Western lifestyles) [71,72].

Thus, innovative techniques in the field of precision medicine may identify specific strains and related metabolic products for microbial biotherapy, ideally tailored for a single patient, such as in personalized medicine. However, the hardly sustainable costs, clustered by age and Westernized or non-Westernized lifestyles, have to be considered [73].

### 3.4. Paracrine- and Endocrine-like Signaling System

All these observations enhance the concept that microbiota not only closely interact with epithelial body tract cells contiguously but that metabolic products derived from bacteria (different depending on the body site and individual and geographic variability) have a key role in local and systemic signaling with other organs. For instance, SCFAs are the most extensively studied microbiota metabolites, and their various effects also include the hematopoietic activity in bone marrow.

Colonic anaerobiosis ensures the growth of a healthy and balanced SCFA-producing microbiota community, dominated by obligate anaerobic bacteria of the phyla Firmicutes and Bacteroides. Concurrently, the hypoxic epithelial niche creates an unfavorable microenvironment for the expansion of dysbiotic strains such as the anaerobic phylum Proteobacteria (Enterobacteriaceae), which identifies gut dysbiosis. Growing evidence supports the system-wide role of SCFAs in dampening inflammation and immunomodulating across the gut–lung axis [74].

SCFAs are also able to generate an extrathymic peripheral Treg cell pool, linked to dampening allergic airway diseases. However, airway bacteria do not produce SCFAs in substantial amounts, likely due to the absence of substrates [45].

Recent research has provided several paths of action through which gut-derived SCFAs are able to power down allergic airway inflammation.

Circulatory *acetate* and *propionate* modulate dendritic cell (DC) hematopoiesis and functionality in the bone marrow during Th2 cell-mediated allergic airway inflammation. These DC precursor cells subsequently populate the lungs, where they mature into CD11b + DCs that are inefficient in allergen presentation and consequently inactivate Th2 cascade effector cells [70].

Thus, the microbiological machinery is equipped with a remote and multichannel communications system called axes: the microbiota–gut–brain axis, the microbiota–immune system axis, the microbiota–bone marrow axis, and the microbiota–gut–lung axis. Virtually, a bidirectional communication could exist between any body site microflora and any body organ, including the upper airways.

For instance, the inferior turbinates, persistently hypertrophic in chronic AR patients, present peculiar epithelial phenotypes resulting in an abnormal niche that may affect the commensal microbiota. Increased epithelial permeability, the stretching of intercellular tight junctions, cytokine secretion, increased mucus secretion, and the local proinflammatory switch of the immune system altering the physiology at the disease site create a bias in local microbiota equilibrium [75].

The mechanism through which the healthy or dysbiotic microbiota impacts a body district’s health or disease and vice versa is only starting to be uncovered and represents a real challenge considering the unpredictable variables that can reroute colonization. In our opinion, it is possible to bypass this challenge, inherent in the probiotic treatment of allergic rhinitis that we will shortly analyze, through the pharmacological use of microbial metabolites, the so-called postbiotics, or through parabiotics, such as probiotic cell components or crude cell extracts.

## 4. Dysbiosis in Allergic Rhinitis: Achievements and Failures of Probiotic Add-On Therapy

The undisputed beneficial effect of probiotics, mainly on the intestinal apparatus, has gained great attention among the scientific community and the general population, rendering probiotics widely used, regardless of a specific therapeutic indication, as an expression of a healthy lifestyle and proper eating habits.

Beginning with gastrointestinal diseases, they are employed as add-on therapeutic options for extraintestinal ones, including allergic disorders.

The rationale for probiotic administration in respiratory allergies is based on mechanistic evidence that we have only partially explained thus far, whereas the beneficial effects on AR are essentially based on clinical proof [45]. 

Probiotics’ definition is “live microorganisms that, when administered in adequate amounts, confer a health benefit on the host” [76]. The most studied and employed probiotic strains are the Lactobacillus and Bifidobacterium genera. Probiotics also include species of the genera Streptococcus, Bacillus, and Enterococcus and the yeast Saccharomyces, which has been used as a probiotic for many years [77].

According to their definition, probiotics are microorganisms that are orally administered, and in an appropriate dose, they have a positive influence on respiratory allergies via the gut–lung axis. Generally, the oral supplementation of probiotic strains seems to be promising; nevertheless, it still has many controversies [45]. 

To summarize, probiotics seem to be promising in the process of immunomodulation by:Increasing the Th1:Th2 ratio, accordingly augmenting the production of the Th1 cytokines and consequently decreasing Th2 cytokine production;Decreasing eosinophil and lymphocyte infiltration in the respiratory tract and allergen-specific IgE production and, conversely, increasing allergen-specific IgG1 and IgG2a, production.Increasing butyrate and secretory IgA production [77,78].

Some recent systematic reviews and meta-analyses [79,80,81,82] have evaluated the efficacy and safety of probiotics in managing AR. Probiotics can play an important role in the prevention and treatment of allergic rhinitis. The clinical benefit of probiotic therapy depends on numerous factors, including the type of bacterium, route of administration, dosage, regimen, and other underlying host factors. Concerning the treatment of AR with probiotics, these studies showed a clinical improvement in the quality of life (QoL) and rhinitis symptoms, sometimes associated with immunological improvement [79,80].

However, a high degree of heterogeneity was observed in most clinical parameters, such as nasal and ocular symptom scores (SS), daily total SS, and QoL. Particularly, nasal stuffiness, rhinorrhea, and nasal itching scores were significantly decreased in the probiotic group compared to the placebo. Sneezing tended to be lower in the probiotic group compared to the placebo. These data are surely intriguing [79]. In addition, the prospective of employing probiotics as adjuvants for allergic immune therapy (AIT) is intriguing and supported by some interesting studies [83]: in children with AR, the addition of probiotics to sublingual immune therapy (SLIT) improved symptom scores and the induction of T regulatory cells after 5 months of treatment [84]; Clostridium butyricum, coadministration with AIT, was shown to improve the efficacy of AIT in patients with AR and asthma [85]; patients who were administered AIT and probiotics showed a reduction in symptom scores with a significant improvement in medication scores after 2 and 4 months of treatment [86]. Therefore, all the interventions aimed at reestablishing, supporting, and preserving the microbiota may constitute a new therapeutic opportunity. There are, however, some concerns. 

Despite these numerous health benefits, surveys of probiotics have highlighted some limitations. They include unknown molecular mechanisms; strain-specific behaviors; the short-lived and niche action of probiotics (allochthonous or autochthonous); the development of antibiotic resistance; the transfer of virulence genes, an obstacle to the colonization of the commensal intestinal microflora; and, not least, the capacity to cause opportunistic infections through bacterial translocation and bacteremia in immunocompromised individuals [77]. Above all, the low concentrations of probiotic-derived metabolites found in specific target sites in the course of the traditional application of live probiotic microorganisms (live biotherapeutics) have been found to be ineffective under in vivo conditions [77].

The genetic background is a pivotal determinant of probiotic efficacy [87], but it must not be unified in human trials, as in cellular or murine models. In addition, available data show that the influence of both organs is reciprocal, the intestinal microbiota affects the respiratory system, and the respiratory microbiota actively affects the intestines [88,89,90]. The vast majority of probiotics that are orally administered are subject to digestive juices which may affect their functionality. An altered microbiota of the respiratory system influences the pathogenesis of respiratory allergy [91]; therefore, the nasal administration of probiotics can be greatly beneficial. However, there are some limitations. The allergic patient may have dysbiosis due to the interference of genetic and environmental factors, which could create an unfavorable niche for sufficient colonization. Therefore, permanent advantages are will not be achieved, and the immunomodulatory intended effect is not assured because the engraftment might not be sufficient, not only in improving symptoms but also, perhaps, in reversing the allergic inflammation.

Consequently, one could perhaps speculate that allergic subjects may present genetic and immunologic conditions that make probiotic rooting more difficult. This hypothesis might assume the need to identify specific strains suitable for allergic subjects and to adjust the duration of supplementation. Therefore, future basic and human studies should address these unmet needs. We currently lack an integrated body biogeography view of the microbial communities in health and disease and during the different stages of life [22].

How bacterial diversity is generated between the different body habitats and how it modulates over time remains to be determined, as well as the existence of predictable biogeographic models capable of being previously modified by altering the natural course of diseases in progress.

Atopy may alter microenvironmental characteristics by playing a stronger role in shaping bacterial communities contributing to the native microbiota selection and, when the allergic rhinitis is overt, by altering the growth or colonization of specific taxa at the expense of others. It is important to evaluate not only the strains of members of the microbiota but also their potential functions in metabolic pathways when considering a therapeutic or preventive intervention for the microbiota.

For example, it may be useful to project a microbial integration based on the body habitat, such as the nose and its biotic conditions.

This concept may have a variety of implications both in therapeutics and prevention. 

The use of probiotics to prevent allergic disease also relates to their utilization by mothers during pregnancy and breastfeeding and the use of probiotics in infants [92,93]. In summary, current evidence does not support the administration of probiotics to prevent any form of allergic disease, except for eczema in high-risk infants (WAO recommendation), but it does favor probiotic supplementation in pregnant/lactating women or infants with a family history of allergic disease [94].

Another interesting topic concerns the prevention of complications: AR affects not only the nose, but in children, it may cause numerous physical complications including otitis media with effusion, chronic sinusitis, and asthma. Poor sleep, poor school performance, hyperactivity, and decreased quality of life lead the list of mental complications that have been highlighted in these patients [95]. Some of these complications may benefit from probiotic treatment or could possibly be prevented. For example, evidence suggests that AR is an inflammatory trigger or an exacerbating condition for CRS. Among pediatric patients with CRS, 36–60% have been diagnosed with AR, and the patients who underwent functional endoscopic sinus surgery took a significantly longer time to recover from the surgery if they had a history of AR. In addition, the symptom scores of CRS patients with AR are significantly improved in patients treated with immunotherapy compared to control patients [96]. In patients with CRS, the microbiome has a reduced bacterial diversity but a higher bacterial load [97]. Furthermore, less stable strains replace more stable bacterial species, favoring the colonization of potentially pathogenic ones, thus resulting in dysbiosis. This process could cause increased permeability in the epithelial barrier for pathogens, leading to the release of inflammatory mediators and consequent chronic inflammation [98]. Probiotics may inhibit the adhesion of pathogens to the mucous barrier, the stabilization of tight junctions in the epithelial layer with a reduction in the permeability of the mucosa, the competitive inhibition of pathogens, the modulation of the immune system, and the production of various substances toxic to pathogenic microorganisms [99]. A recent review described how probiotics may restore the tight and adherens junctions of the epithelial barrier and the host’s immune modulation via interaction with dendritic cells. This enhances Tregs and downregulates T-helper 1 and T-helper 2. Thus, probiotics, through these mechanisms, may prevent CRS in patients with AR and may mitigate CRS and AR in patients suffering from both diseases [100]. However, research in this area is fervently progressing in order to provide a better knowledge of optimal strains, dosages, timing, and duration of probiotic administration in the prevention of allergic disease. 

## 5. Last Advances and News Expectations: Postbiotics and Parabiotics

The considerable efforts that have been made, and the encouraging results obtained, from studying probiotics have opened new doors that allow us to innovate the point of view in the field of microbial biotherapy. It was recognized that some mechanisms underlying the beneficial effects of probiotics are not strictly dependent on live microorganisms. Since then, postbiotic and parabiotic notions have been developed as new categories of compounds able to directly exert the biological responses typical of the healthy microbiota [101]. 

Even though no formalized definitions are currently available, the most commonly employed for postbiotics is “non-viable bacterial products or metabolic products from microorganisms that have biological activity in the host”, and for parabiotics, it is “ghost or inactivated probiotics” or, better yet, “non-viable microbial cells (either intact or broken) or crude cell extracts which when administered (either orally or topically) in adequate amounts, confer a benefit on the human or animal consumer” [77]. 

The various postbiotic molecules include, for instance, vitamins and flavonoids, organic acids, SCFAs, secreted proteins and amino acids, bacteriocins, neurotransmitters, secreted biosurfactants, terpenoids, and many others [77,78]. Parabiotics, simplistically, are the intact, inactivated microbial cells or cell lysates of probiotics containing cell components such as teichoic acids, peptidoglycan-derived muropeptides, pili, fimbriae, flagella, polysaccharides teichoic acids, and more [77,78].

The known molecular structure and, consequently, the predictable and specific downstream pathway it elicits, in addition to the absence of the risk of bacterial translocation and the acquisition of antibiotic resistance, represent the main strengths of postbiotic and parabiotic utilization.

The different postbiotic and parabiotic molecules exert several effects on many diseases such as obesity, hypertension, coronary artery diseases, and cancer, mainly controlling inflammation, oxidative stress, and immune modulation. 

These functions explain the great interest demonstrated in the potentiality of postbiotics and parabiotics in improving host physiology, contributing to the prevention and treatment of many diseases and preparing the niche for the subsequent engraftment of probiotics, assuming a powerful combined approach to restore a more permanent equilibrium. 

Specifically, in our field of interest (namely, allergic rhinitis), as already described, with respect to SCFAs, various research has highlighted that certain metabolites secreted by specific strains act on the differentiation and function of CD4 + T cells influencing allergic inflammation. 

Further examples consist of bile acid-derived metabolites, microbial polysaccharides, indole-3-lactic acid (IDO), 12,13-diHOME, and the already known effects of B6 and B3 vitamins on Treg cells [5].

3β-hydroxy deoxycholic acid, a bacterial transformation product of bile acids in the colon, modulates DC function, enhances differentiation, and increases the amount of RORγt + pTregs of the intestinal mucosa, regulating type 2 inflammation [102]. Microbial polysaccharides, acting through Toll-like receptor-2 (TLR2) signaling, have a role both in Treg induction and in increasing the local production of anti-inflammatory IL-10 [103,104]. Indole-3-lactic acid (IDO), a tryptophan metabolite produced by both L. reuteri [105] and B. infantis, upregulates galectin-1, which inhibits both Th2 and Th17 in human studies [106]. In addition, in a longitudinal study that involved multi-sensitizing atopic children for 3 years—in which gut microbiota dysbiosis and metabolic activity were studied starting at 3 months of age—before developing atopic disorders, fecal metabolites induced an increase in IL-4-secreting Th2 cells with a Treg suppressive effect. 12,13-diHOME, a linoleic acid metabolite, isolated in these children, was observed to amplify lung inflammation in asthmatic mice and to be increased at 1 month of age in infants, who developed atopy by age two [102].

Concluding this exemplifying overview, we need human clinical trials focusing on the efficacy and safety of these compounds in AR in adults and children.

## 6. Conclusions

Allergic rhinitis is a typical type 2 disease characterized by a polarization of innate and adaptive type 2 cells (ILC2 and Th2) and a functional and allergen-specific defect of Tregs. This immune derangement is also associated with a deficient type 1 immune response. This immunologic profile is mutually linked to dysbiosis and impaired biodiversity, affecting both airways and the digestive tract. These considerations suggest the possibility of restoring a physiologic and tolerogenic response to allergens, replacing the defective intestinal microbiota and, hopefully, also the respiratory one. The existence of a gut–airway axis supports this eventuality, that is, the crosstalk between immunity and microbiota in the gut and airways. The ideal candidate to manipulate both microbiota and immunity was identified in probiotics. This hypothesis was tested by a series of in vitro and in vivo studies aiming to demonstrate the plastic rebuilding of the immune system with probiotics. The proof of concept was the probiotics’ capability of restoring dysbiosis and rebalancing immunity to clinical and immunological tolerance. However, the use of probiotics still generates some perplexities. Namely, the studies on the use of probiotics have not yet produced a confident and convincing demonstration of their actual ability to modify the type 2 response. In addition, concerns about complete safety have also been raised. Pending robust and definitive evidence, the current thought on the possible use of postbiotics and parabiotics is open. At present, this avenue remains theoretical and to be explored, but it will certainly be a fascinating path to follow. Meanwhile, there is a need to conduct new studies on probiotics following proper and appropriate methodology to address unmet needs.

Probably, as in other medical specialties, we will hear more and more often in the future about postbiotics and parabiotics in allergology, and their regular consumption, as with probiotics, may represent a huge advantage in terms of proactive prevention.

Today, we have the knowledge to develop a functional and structured approach aimed at the treatment of specific diseases using what already happens naturally. We should encourage it, considering that this peaceful coexistence has been associated with mutual benefits since the birth of human beings.

## Data Availability

Not applicable.

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
