# Peer review of "(R)Evolution in Allergic Rhinitis Add-On Therapy: From Probiotics to Postbiotics and Parabiotics"

_jcm, 2022, doi:10.3390/jcm11175154_

Round 1

Reviewer 1 Report

The review article is informative and well presented.   A few comments for authors' consideration : 

1. please justify with more references to support "several studies proposed a rearrangement of infant microbiota mainly determined by body niches and not by the delivery type" line 93/94.   only 1 small study was quoted [ reference 19] . 

2. the term " paracrine and endocrine" interactions (section 3.4) may be used as a borrowed term here but a bit confusing as the terms are more commonly used in endocrinology 

3. line 316-318 about efficacy of probiotics and AR management , please elaborate on efficacy of treatment and prevention separately 

Author Response

  1. Please justify with more references to support "several studies proposed a rearrangement of infant microbiota mainly determined by body niches and not by the delivery type" line 93/94.   Only 1 small study was quoted [ reference 19] .

Dear reviewer,

Thank you very much for the suggestions. We have modified the manuscript according your comment in greater depth and with a clearer presentation of the argument expanding our literature search and adding mor references.

 From line 94 to 129:

 “Previous studies have demonstrated that in adult individuals not only each body site has different and unique microbial communities [19]–[22] but also contiguous sites (e.g. supragingival plaque vs. tongue) have significant variations in microbiota composition. probably  even the smallest of variations in body microniches conditions may influence the colonization[19], [23]. Also, a study on preterm infants with very low birth weight (LBW) demonstrated that the microbiota of skin, saliva, and stool rapidly diverged (after the first 15 days of life) within the first three weeks of life: body site was the primary determinant of bacterial community composition in the LBW infants[24]. More recently, an in-depth exploration of infant microbiome maturation highlights that at the time of delivery, the microbiota comprises some taxa of the maternal skin and vaginal microbiota, while at 6 weeks of life the main patterns of community variation are associated with the body site and the selection is driven by the body niche asset. The infant stool, nares, and oral cavity harbored microbial communities distinct from the parental ones, reflecting age-related physiological differences between the maternal and infant body habitats, including nutrient availability and oxygen exposure[23]. In recent years, the impact delivery, vaginal or cesarean, on the infant microbiota has been subjected to a critical revision also because the clinical decision to deliver via cesarean often implies an underlying maternal or fetal morbidity frequently accompanied by medications, including antibiotics and anti-inflammatory and analgesics, all factors together significant confounders[25]. In addition, concurrent studies characterized bacteria mapped to the placenta and amniotic fluid of preterm and healthy, term pregnancies in humans[26]–[29], and maternal transmission of bacteria to the fetal gut during gestation in mice, consistent with microbial colonization of the mouse fetus occurring before delivery[30]. Also the meconium has been shown to harbor a microbial community similar to that of the amniotic fluid and placenta[31]–[33] that vary with maternal glycemic control, but not with delivery modality[34]. Thus,  even if additional studies are needed to evaluate potential mechanisms of transmission and its potential impact on fetal microbiota long-term programming, several studies have indicated that the early composition of the gut may be influenced by maternal diet or health status ( i.e. pre-pregnancy obesity, and maternal glycemic control) in the last trimester of pregnancy, independent of the mode of delivery[35].  Notably, these maternal health conditions also increase the risk that the pregnancy will be delivered by cesarean delivery[36].  Therefore, the predominant role of delivery modality determining microbiota colonization must be assessed considering the complexity of the clinical setting leading to the choice of delivery mode.”

  1. the term " paracrine and endocrine" interactions (section 3.4) may be used as a borrowed term here but a bit confusing as the terms are more commonly used in endocrinology 

Dear reviewer,

it was our intention to borrow the relative term "paracrine and endocrine" to underline the endocrinological-like functions of microbial metabolites capable of modulating the metabolic activities of other bacterial and human cells near and in other body sites of the organism. However, we recognize, as you suggested, that the use of this terminology could generate confusion in reading the title of the paragraph, but also the curiosity necessary to continue reading to better understand what we mean. However, taking into account your advice, we thought of modifying the paragraph title  in “Paracrine- and Endocrine-like Signaling System” (Line 286).

  1. Line 316-318 about efficacy of probiotics and AR management , please elaborate on efficacy of treatment and prevention separately.

Dear reviewer, given the length of this paper, we had thought of summarizing the topic, also considering that the references cited are very detailed reviews and meta-analyzes on the topic of prevention and treatment of RA with probiotics. Therefore expanding the argument further may be superfluous. However, in summarizing the topic, as you noted, we better argued the effects of the treatment rather than on the attempts to prevent allergic diseases.             Following your advice, we better elaborate the topic specifying separately what concerns treatment and what concern prevention:

  • Line 363-366: “Probiotics can play an important role in the prevention and treatment of allergic rhinitis. The clinical benefit of probiotic therapy depends on numerous factors, including type of bacterium, route of administration, dosage, regimen, and other underlying host factors. Concerning the treatment of AR with probiotics..”
  • Line 375-383: “Also, the prospective of employing probiotics as adjuvants for Allergic Immune Therapy (AIT) is intriguing and supported by some interesting studies[83]: in children with AR, the addition of probiotics to Sub Lingual Immune Therapy (SLIT) improved symptom scores and the induction of T regulatory cells after 5 months of treatment [84]; Clostridium butyricum coadministration with AIT shown to improve the efficacy of AIT in patients with AR and asthma[85]; patients who were administered AIT and probiotics showed a reduction of symptoms scores with a significant improvement in medication scores after 2 and 4 months of treatment [86].”
  • Line 411- 431 :“We currently lack an integrated body biogeography view of the microbial communities in health, disease and during the different stages of life[22].

How bacterial diversity is generated between the different body habitats and how it modulates over time remains to be determined, as well as the existence of predictable biogeographic models capable of being previously modified by altering the natural course of diseases in progress. Atopy may alter the microenvironmental characteristics playing a stronger role in shaping bacterial communities contributing to the native microbiota selection, and when the Allergic Rhinitis is overt, by altering the growth or colonization of specific taxa at the expense of others. It is important to evaluate not only the strains of members of the microbiota but their potential functions on metabolic pathways when considering a therapeutic or preventive intervention for the microbiota. For example, may be useful to project a microbial integration based on the body habitat, such as the nose and its biotic conditions.

This concept may have a variety of implications both in therapeutics and prevention.

The use of probiotics to prevent allergic disease concerns their utilization by mothers during pregnancy and breastfeeding and the use of probiotics in infants[92], [93]. In summarizing, current evidence does not support the administration of probiotics to prevent any form of allergic disease, with except for eczema in high-risk infants (WAO recommendation), but does favor probiotic supplementation in pregnant/lactating women and infants with a family history of allergic disease[94].”

  1. please justify with more references to support "several studies proposed a rearrangement of infant microbiota mainly determined by body niches and not by the delivery type" line 93/94.   only 1 small study was quoted [ reference 19] . 

Dear reviewer,

Thank you very much for the suggestions. We have modified the manuscript according your comment in greater depth and with a clearer presentation of the argument expanding our literature search and adding mor references.

 From line 94 to 129:

 “Previous studies have demonstrated that in adult individuals not only each body site has different and unique microbial communities [19]–[22] but also contiguous sites (e.g. supragingival plaque vs. tongue) have significant variations in microbiota composition. probably  even the smallest of variations in body microniches conditions may influence the colonization[19], [23]. Also, a study on preterm infants with very low birth weight (LBW) demonstrated that the microbiota of skin, saliva, and stool rapidly diverged (after the first 15 days of life) within the first three weeks of life: body site was the primary determinant of bacterial community composition in the LBW infants[24]. More recently, an in-depth exploration of infant microbiome maturation highlights that at the time of delivery, the microbiota comprises some taxa of the maternal skin and vaginal microbiota, while at 6 weeks of life the main patterns of community variation are associated with the body site and the selection is driven by the body niche asset. The infant stool, nares, and oral cavity harbored microbial communities distinct from the parental ones, reflecting age-related physiological differences between the maternal and infant body habitats, including nutrient availability and oxygen exposure[23]. In recent years, the impact delivery, vaginal or cesarean, on the infant microbiota has been subjected to a critical revision also because the clinical decision to deliver via cesarean often implies an underlying maternal or fetal morbidity frequently accompanied by medications, including antibiotics and anti-inflammatory and analgesics, all factors together significant confounders[25]. In addition, concurrent studies characterized bacteria mapped to the placenta and amniotic fluid of preterm and healthy, term pregnancies in humans[26]–[29], and maternal transmission of bacteria to the fetal gut during gestation in mice, consistent with microbial colonization of the mouse fetus occurring before delivery[30]. Also the meconium has been shown to harbor a microbial community similar to that of the amniotic fluid and placenta[31]–[33] that vary with maternal glycemic control, but not with delivery modality[34]. Thus,  even if additional studies are needed to evaluate potential mechanisms of transmission and its potential impact on fetal microbiota long-term programming, several studies have indicated that the early composition of the gut may be influenced by maternal diet or health status ( i.e. pre-pregnancy obesity, and maternal glycemic control) in the last trimester of pregnancy, independent of the mode of delivery[35].  Notably, these maternal health conditions also increase the risk that the pregnancy will be delivered by cesarean delivery[36].  Therefore, the predominant role of delivery modality determining microbiota colonization must be assessed considering the complexity of the clinical setting leading to the choice of delivery mode.”

  1. the term " paracrine and endocrine" interactions (section 3.4) may be used as a borrowed term here but a bit confusing as the terms are more commonly used in endocrinology 

Dear reviewer,

it was our intention to borrow the relative term "paracrine and endocrine" to underline the endocrinological-like functions of microbial metabolites capable of modulating the metabolic activities of other bacterial and human cells near and in other body sites of the organism. However, we recognize, as you suggested, that the use of this terminology could generate confusion in reading the title of the paragraph, but also the curiosity necessary to continue reading to better understand what we mean. However, taking into account your advice, we thought of modifying the paragraph title  in “Paracrine- and Endocrine-like Signaling System” (Line 286)

  1. line 316-318 about efficacy of probiotics and AR management , please elaborate on efficacy of treatment and prevention separately 

Dear reviewer, given the length of this paper, we had thought of summarizing the topic, also considering that the references cited are very detailed reviews and meta-analyzes on the topic of prevention and treatment of RA with probiotics. Therefore expanding the argument further may be superfluous. However, in summarizing the topic, as you noted, we better argued the effects of the treatment rather than on the attempts to prevent allergic diseases.             Following your advice, we better elaborate the topic specifying separately what concerns treatment and what concern prevention:

  • Line 363-366: “Probiotics can play an important role in the prevention and treatment of allergic rhinitis. The clinical benefit of probiotic therapy depends on numerous factors, including type of bacterium, route of administration, dosage, regimen, and other underlying host factors. Concerning the treatment of AR with probiotics..”
  • Line 375-383: “Also, the prospective of employing probiotics as adjuvants for Allergic Immune Therapy (AIT) is intriguing and supported by some interesting studies[83]: in children with AR, the addition of probiotics to Sub Lingual Immune Therapy (SLIT) improved symptom scores and the induction of T regulatory cells after 5 months of treatment [84]; Clostridium butyricum coadministration with AIT shown to improve the efficacy of AIT in patients with AR and asthma[85]; patients who were administered AIT and probiotics showed a reduction of symptoms scores with a significant improvement in medication scores after 2 and 4 months of treatment [86].”
  • Line 411- 431 :“We currently lack an integrated body biogeography view of the microbial communities in health, disease and during the different stages of life[22].

How bacterial diversity is generated between the different body habitats and how it modulates over time remains to be determined, as well as the existence of predictable biogeographic models capable of being previously modified by altering the natural course of diseases in progress. Atopy may alter the microenvironmental characteristics playing a stronger role in shaping bacterial communities contributing to the native microbiota selection, and when the Allergic Rhinitis is overt, by altering the growth or colonization of specific taxa at the expense of others. It is important to evaluate not only the strains of members of the microbiota but their potential functions on metabolic pathways when considering a therapeutic or preventive intervention for the microbiota. For example, may be useful to project a microbial integration based on the body habitat, such as the nose and its biotic conditions.

This concept may have a variety of implications both in therapeutics and prevention.

The use of probiotics to prevent allergic disease concerns their utilization by mothers during pregnancy and breastfeeding and the use of probiotics in infants[92], [93]. In summarizing, current evidence does not support the administration of probiotics to prevent any form of allergic disease, with except for eczema in high-risk infants (WAO recommendation), but does favor probiotic supplementation in pregnant/lactating women and infants with a family history of allergic disease [94].”

Reviewer 2 Report

this is a very detailed review regarding the microbiome in allergic rhinitis. 

I think that adding an information on the role of the microbiome in chronic rhinosinusitis with or without nasal polyposis will contribue to this manuscript. There are studies demonstareting a change in the microbiome in patients with reccurent disease after a surgery in compare to patients with non reccurent disease .

Second ,  are there any practical sugestions for treatment in atopic patients following this detaled review?  

Author Response

This is a very detailed review regarding the microbiome in allergic rhinitis. 

I think that adding an information on the role of the microbiome in chronic rhinosinusitis with or without nasal polyposis will contribute to this manuscript. There are studies demonstareting a change in the microbiome in patients with reccurent disease after a surgery in compared to patients with non reccurent disease.

Dear reviewer,

Thank you for the suggestions. We have modified the manuscript according your comment expanding our text to deepen the role of the microbiome in chronic rhinosinusitis:

Line 432-459 :  “Another interesting topic concerns the prevention of complications: AR affected not only the nose, but in children may have numerous physical complications including otitis media with effusion, chronic sinusitis, and asthma. Sleep, poor school performance, hyperactivity, and decreased quality of life lead the list of mental complications that have been highlighted in these patients[95]. Some of these complications may benefit from probiotic treatment or could possibly be prevented. For example, several evidence suggests that AR is an inflammatory trigger or an exacerbating condition for CRS. Among pediatric patients with CRS, 36–60% have been diagnosed with AR , and the patients who underwent functional endoscopic sinus surgery takes significantly longer time to recover from the surgery if they had a history of AR. In addition  the symptoms scores of CRS patients with AR are significantly improved in patients treated with immunotherapy compared to control patients[96]. In patients with CRS, the microbiome has a reduced bacterial diversity, but a higher bacterial load[97]. Furthermore, less stable strains replace more stable bacterial species favoring the colonization of potentially pathogenic ones resulting in dysbiosis. This process could cause increased permeability of the epithelial barrier to pathogens leading to the release of inflammatory mediators and consequent chronic inflammation[98]. Probiotics may inhibit the adhesion of pathogens to the mucous barrier, the stabilization of tight junctions in the epithelial layer with a reduction in the permeability of the mucosa, the competitive inhibition of pathogens, modulation of the immune system, and the production of various substances toxic to pathogenic microorganisms[99]. A recent review described how probiotics may restore the tight and adherence junctions of the epithelial barrier and the host’s immune modulation via interaction with dendritic cells. This enhances Tregs and downregulates T-helper 1 and T-helper 2. Thus, probiotics through these mechanisms may prevent CRS in patients with AR and may mitigate CRS and AR in patients suffering from both the disease[100]. However, research in this area is in fervent progress to provide a better knowledge of optimal strains, dosages, timing, and duration of probiotic administration in the prevention of allergic disease.”

Second, are there any practical suggestions for treatment in atopic patients following this detailed review?  

At present, practicing clinicians can avail themselves of intestinal flora modulators as add-on therapy in the treatment of allergic diseases. Their effects on the treatment of allergic diseases remain controversial. In each allergic disease - allergic rhinitis; allergic asthma; atopic dermatitis; food allergy - there are substantial uncertainties on the efficacy of gut microflora modulation in prevention and treatment. Also, the official guidelines do not recommend the administration of probiotics to prevent allergic disease, but in practice, their administration in case of overt pathology may be useful not only empirically. In our opinion, the pharmacological use of microbiota metabolic derivates may bypass the difficulty related to probiotic administration, but this innovative approach requires extensive research. Our suggestions are directed more to the scientific community by changing the point of view through which we look at the microbiota: a complex system that is metabolically active and potentially interconnected with every human and bacterial cell of our organism.